# Asynchronous and Slow-Wave Oscillatory States in Connectome-Based Models of Mouse, Monkey and Human Cerebral Cortex

Maria Sacha [1,†], Jennifer S. Goldman [1,†], Lionel Kusch [2] and Alain Destexhe [1,*]

1  CNRS, Paris-Saclay Institute of Neuroscience (NeuroPSI), Paris-Saclay University, 91400 Saclay, France; jennifer.goldman@mail.mcgill.ca (J.S.G.)
2  Institut de Neurosciences des Systèmes, Aix-Marseille University, INSERM, 13009 Marseille, France
*  Correspondence: alain.destexhe@cnrs.fr
†  These authors contributed equally to this work.

**Abstract:** Thanks to the availability of connectome data that map connectivity between multiple brain areas, it is now possible to build models of whole-brain activity. At the same time, advances in mean-field techniques have led to biologically based population models that integrate biophysical features such as membrane conductances or synaptic conductances. In this paper, we show that this approach can be used in brain-wide models of mice, macaques, and humans.We illustrate this approach by showing the transition from wakefulness to sleep, simulated using multi-scale models, in the three species. We compare the level of synchrony between the three species and find that the mouse brain displays a higher overall synchrony of slow waves compared to monkey and human brains. We show that these differences are due to the different delays in axonal signal propagation between regions associated with brain size differences between the species. We also make the program code—which provides a set of open-source tools for simulating large-scale activity in the cerebral cortex of mice, monkeys, and humans—publicly available.

**Keywords:** cerebral cortex; asynchronous states; slow oscillations; sleep; computational models; mean-field models





## 1. Introduction

Brain activity can display widely different states, ranging from the two extremes of asynchronous activity, typical of the aroused brain, and slow and synchronized oscillatory states, reminiscent of slow-wave sleep or anesthesia [1,2]. To model such "macroscopic" states of brain activity, one should ideally take into account "microscopic" features that are key to the genesis of brain states. Therefore, models that can account for the biophysical features of brain states must be multi-scale. At the microscopic scale, the genesis of neural activity depends on membrane conductances, such as those responsible for spike-frequency adaptation, which regulate transitions between activity states in cortical slices [3]. The macroscopic features of neural activity between brain states may also crucially depend on recurrent connectivity between excitatory and inhibitory neuron types in the cerebral cortex. Computational models have been successful at simulating such activity states, either the asynchronous irregular state [3–6] or the slow-wave state with "Up" and "Down" state dynamics [3,7,8]. Few models have been able to simulate both states, however, because their transition depends on the presence of spike-frequency adaptation conductances. These transitions can be modeled either by Hodgkin–Huxley type models [3] or by integrate-and-fire type models that include spike-frequency adaptation, such as the adaptive exponential (AdEx) model [8].

Although the genesis of asynchronous and slow oscillatory states can be well modeled at the local network scale, as this is typically based on in vitro activity [9], it is less clear how

such activity is organized at larger scales. Here, the activities generated at the local scale in each brain area interact through long-range inter-areal connectivity. The precise pattern of inter-areal connectivity is provided by the connectome, an area of interest on which many open-access studies have been published for a number of species, such as mice [10], macaque monkeys [11], and humans [12]. Thus, the information is available to construct large-scale models that will include all brain areas and their connectivity. However, simulating such models at the cellular scale represents a huge investment of computational resources and is out of reach for most researchers with no access to such resources.

An alternative approach is to simulate brain activity using population models, which are much less demanding on computational resources. However, such models need to contain enough biological realism to include the relevant biophysical mechanisms necessary to generate brain states, as described above, such as membrane conductances and synaptic receptor types. Here, we use an approach that was provided in a recent series of papers [6,8,13,14]. A mean-field model was first derived for AdEx spiking networks [6,8] and it successfully generated both asynchronous and synchronized slow-wave dynamics, as well as their transitions, by controlling spike-frequency adaptation, as in previous experiments. This AdEx mean-field model is much faster at simulation than the spiking networks. Next, such mean-field models were integrated into large-scale models of the human brain [13,14] and could successfully simulate asynchronous states, similar to wakefulness, and slow-wave oscillatory states, similar to slow-wave sleep, as well as their responsiveness to external stimuli (for a similar approach using current-based models, see [15]).

In the present paper, we extend this approach to two species: mice and macaque monkeys. We show the integration of the AdEx mean-field models into large-scale models of the entire cerebral cortex, comparing three species: mice, monkeys, and humans. In particular, we illustrate simulated brain dynamics for asynchronous and slow-wave activity across the whole cerebral cortex.

## 2. Materials and Methods

As schematized in Figure 1, we used three types of models: a network of spiking neurons, a mean-field model of this network, and a network of mean-field models implemented in The Virtual Brain (TVB). In this section, we describe these models successively.

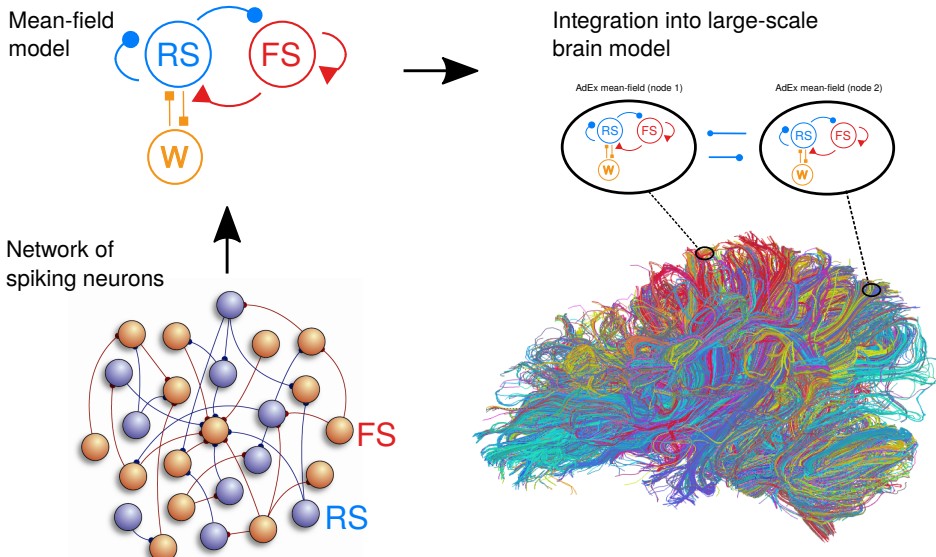

**Figure 1.** Scheme of the bottom-up construction of large-scale brain models. Starting from a spiking neuronal network (bottom left), a mean-field model (top right) is derived. This mean-field model represents the activity of every node in a large-scale model (right), where the connectivity is specified by the connectome (shown here for the human brain). Right panel, adapted from [14].

*2.1. Spiking Network Model*

We considered networks of integrate-and-fire neuron models displaying spike-frequency adaptation, based on two previous papers [6,16]. We used the Adaptive Exponential (AdEx) integrate-and-fire model [17]. We considered a population of $N = 10^4$ neurons, randomly connected with a connection probability of $p = 5\%$. We considered excitatory and inhibitory neurons, with 20% inhibitory neurons. The AdEx model allows us to define two cell types, "regular-spiking" (RS) excitatory cells, displaying spike-frequency adaptation, and "fast spiking" (FS) inhibitory cells, with no adaptation. The dynamics of these neurons is given by the following equations:

$$
\begin{aligned}
c_m \frac{dv_k}{dt} &= g_L(E_L - v_k) + \Delta_k e^{\frac{v_k - v_{thr}}{\Delta_k}} - w_k + I_{syn} \\
\tau_w \frac{dw_k}{dt} &= -w_k + b \sum_{t_{sp}(k)} \delta(t - t_{sp}(k)) \\
&\quad + a(v_k - E_L),
\end{aligned}
\tag{1}
$$

where $c_m = 200$ pF is the membrane capacitance, $v_k$ is the voltage of neuron $k$, and whenever $v_k > v_{thr} = -50$ mV ($v_{thr}$ being the voltage threshold for the action potential) at time $t_{sp}(k)$, $v_k$ is reset to the resting voltage $v_{rest} = -65$ mV and fixed to that value for a refractory period $T_{refr} = 5$ ms. The leak term $g_L$ has a fixed conductance of $g_L = 10$ nS and the leakage reversal $E_L$ is $-65$ mV. The exponential term $\Delta_k$ has a different strength for RS and FS cells, i.e., $\Delta_e = 2$ mV ($\Delta_i = 0.5$ mV) for excitatory (inhibitory) cells. Parameters $a$ and $b$ describe the adaptation conductance and current increment, respectively. Inhibitory neurons were modeled as fast-spiking FS neurons without adaptation ($a_i = b_i = 0$ for all inhibitory neurons), whereas excitatory regular-spiking RS neurons had a lower level of excitability due to the presence of adaptation (for the RS cells, parameter $b_e$ varied in our simulations, while we fixed $a_e = 4$ nS and the adaptation time constant $\tau_w = 500$ ms, unless otherwise specified). The $\delta$-function indicates that whenever a neuron fires at time $t_{sp}(k)$, the adaptation current $w_k$ is incremented by an amount $b$. The synaptic current $I_{syn}$ received by neuron $i$ is the result of the spiking activity of all neurons $j \in \mathrm{pre}(i)$ pre-synaptic to neuron $i$. This current can be decomposed in the synaptic conductances evoked by excitatory E and inhibitory I pre-synaptic spikes, as follows:

$$
I_{syn} = G^e_{syn}(E_e - v_k) + G^i_{syn}(E_i - v_k),
$$

where $E_e = 0$ mV ($E_i = -80$ mV) is the excitatory (inhibitory) reversal potential. Excitatory synaptic conductances $G^e_{syn}$ were modeled by a decaying exponential function that sharply increases by a fixed amount $Q_e$ at each pre-synaptic spike, that is:

$$
G^e_{syn}(t) = Q_e \sum_{exc.pre} \Theta(t - t^e_{sp}(k)) \, e^{-(t - t^e_{sp}(k))/\tau_e},
$$

where $\Theta$ is the Heaviside function, $\tau_e = \tau_i = 5$ ms is the characteristic decay time of excitatory and inhibitory synaptic conductances, and $Q_e = 1$ nS ($Q_i = 5$ nS) is the excitatory (inhibitory) quantal conductance. Inhibitory synaptic conductances are modeled using the same equation with $e \to i$. This network displays two different states according to the level of adaptation $b_e$ (hereafter indicated as $b$), with $b = 5$ pA for asynchronous irregular states, and $b = 60$ pA for up–down states (Figure 2A; see [6] for details and Table 1 for a synopsis of the model parameters).

**Table 1.** Model parameters.

| Parameter Name | Symbol | Value | Unit |
|---|---|---|---|
| **Cellular Properties** | | | |
| Leak conductance | $g_L$ | 10 | nS |
| Leak reversal potential | $E_L$ | −65 | mV |
| Membrane capacitance | $C_m$ | 200 | pF |
| Resting voltage | $v_{rest}$ | −65 | mV |
| Action Potential threshold | $v_{thr}$ | −50 | mV |
| Refractory period | $T_{refr}$ | 5 | ms |
| Adaptation time constant | $\tau_w$ | 500 | ms |
| *Excitatory Neuron* | | | |
| Spike sharpness | $\Delta_e$ | 2 | mV |
| Adaptation current increment | $b_e$ | varies | pA |
| Adaptation conductance | $a_e$ | 4 | nS |
| *Inhibitory Neuron* | | | |
| Spike sharpness | $\Delta_i$ | 0.5 | mV |
| Adaptation current increment | $b_i$ | 0 | pA |
| Adaptation conductance | $a_i$ | 0 | ns |
| **Synaptic Properties** | | | |
| *Excitatory Neuron* | | | |
| Reversal potential | $E_e$ | 0 | mV |
| Quantal conductance | $Q_e$ | 1 | nS |
| Decay time of synaptic conductance | $\tau_e$ | 5 | ms |
| *Inhibitory Neuron* | | | |
| Reversal potential | $E_i$ | −80 | mV |
| Quantal conductance | $Q_i$ | 5 | nS |
| Decay time of synaptic conductance | $\tau_i$ | 5 | ms |
| **Network Properties** | | | |
| Total network size | $N$ | 10,000 | |
| Connectivity probability | $p$ | 0.05 | |
| Fraction of inhibitory cells | $g_{ei}$ | 0.2 | |

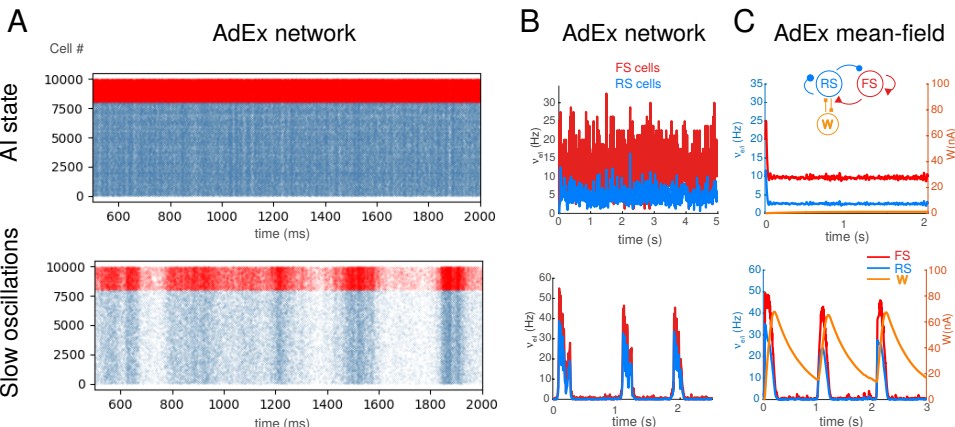

**Figure 2.** Asynchronous irregular and slow oscillatory states in AdEx networks and mean-field models (**A**). Raster plots of excitatory RS (blue) and inhibitory FS (red) AdEx neurons in a network during asynchronous irregular (AI) state (**top**), and slow oscillations with up–down states (**bottom**). The two states differed from the values of adaptation parameter *b* (*b* = 1 pA for AI state, and 20 pA for slow oscillations shown here) (**B**). Corresponding mean firing rates of the two populations (**C**). Mean-field model of AdEx networks with adaptation (scheme in inset). The mean-field model reproduces the two states and the corresponding firing rate variations (adaptation current *W* shown in yellow). Adapted from [8,13].

### 2.2. Mean-Field Models

We considered a population model of a network of AdEx neurons, using a master equation formalism originally developed for balanced networks of integrate-and-fire neurons [18]. This model was adapted to AdEx networks of RS and FS neurons [6], and later modified to include adaptation [8]. The latter version is used here, which corresponds to the following equations:

$$
\begin{aligned}
T\frac{\partial \nu_\mu}{\partial t} &= (F_\mu - \nu_\mu) + \frac{1}{2}c_{\lambda\eta}\frac{\partial^2 F_\mu}{\partial \nu_\lambda \partial \nu_\eta} \\
T\frac{\partial c_{\lambda\eta}}{\partial t} &= \delta_{\lambda\eta}\frac{F_\lambda(1/T - F_\eta)}{N_\lambda} + (F_\lambda - \nu_\lambda)(F_\eta - \nu_\eta) \\
&\quad + \frac{\partial F_\lambda}{\partial \nu_\mu}c_{\eta\mu} + \frac{\partial F_\eta}{\partial \nu_\mu}c_{\lambda\mu} - 2c_{\lambda\eta} \\
\tau_w\frac{\partial W}{\partial t} &= -W + b\tau_w\nu_e + a(\mu_V(\nu_e, \nu_i, W) - E_L),
\end{aligned}
\tag{2}
$$

where *T* is the time constant, $\mu = \{e, i\}$ is the population index (excitatory or inhibitory), $\nu_\mu$ is the population firing rate, and $c_{\lambda\eta}$ is the covariance between populations $\lambda$ and $\eta$. *W* is a population adaptation variable [8]. $\delta_{\lambda\eta} = 1$ if $\lambda = \eta$ and zero otherwise. The function $F_{\mu=\{e,i\}} = F_{\mu=\{e,i\}}(\nu_e, \nu_i, W)$ is the transfer function, which describes the firing rate of population $\mu$ as a function of excitatory and inhibitory inputs (with rates $\nu_e$ and $\nu_i$) and adaptation level *W*. These functions were estimated previously for RS and FS cells and in the presence of adaptation [8].

At the first order, i.e., neglecting the dynamics of the covariance terms $c_{\lambda\eta}$, this model can be written simply as follows:

$$
T\frac{d\nu_\mu}{dt} = (F_\mu - \nu_\mu),
\tag{3}
$$

together with Equation (3). This system is equivalent to the well-known Wilson–Cowan model [19], with the specificity that the functions *F* need to be obtained according to

the specific single-neuron model under consideration. These functions were obtained previously for AdEx models of RS and FS cells [6,8] and the same are used here.

For a cortical volume modeled as two populations of excitatory and inhibitory neurons, the equations (at first order) can be written as follows:

$$
\begin{aligned}
T\frac{d\nu_e}{dt} &= \mathcal{F}_e(\nu_e + \nu_{aff} + \nu_{drive}, \nu_i) - \nu_e \\
T\frac{d\nu_i}{dt} &= \mathcal{F}_i(\nu_e + \nu_{aff}, \nu_i) - \nu_i \\
\tau_w\frac{dW}{dt} &= -W + b\tau_w\nu_e + a(\mu_V(\nu_e, \nu_i, W) - E_L),
\end{aligned}
\tag{4}
$$

where $\nu_{aff}$ is the afferent thalamic input to the population of excitatory and inhibitory neurons and $\nu_{drive}$ is an external noisy drive. The function $\mu_V$ is the average membrane potential of the population and is given by the following equation:

$$
\mu_V = \frac{\mu_{Ge}E_e + \mu_{Gi}E_i + g_L E_L - W}{\mu_{Ge} + \mu_{Gi} + g_L},
$$

where the mean excitatory conductance is $\mu_{Ge} = \nu_e K_e \tau_e Q_e$. Similar to inhibition $K_e = (1 - g_{ei})Np$, $g_{ei}$ is the fraction of inhibitory neurons in the network, $N$ is the total size of the network, and $p$ is the probability of connection.

This system describes the population dynamics of a single isolated cortical column and was shown to closely match the dynamics of the spiking network (Figure 2; [8]).

### 2.3. Networks of Mean-Field Models

Extending our previous work at the mesoscale [6] to model large brain regions, we define networks of mean-field models, representing interconnected cortical columns (each described by a mean-field model). For simplicity, we considered only excitatory interactions between cortical columns, while inhibitory connections remained local to each column. The equations of such a network, expanding the two-population mean-field model (Equation (4)), are given in the following equation:

$$
\begin{aligned}
T\frac{d\nu_e(k)}{dt} &= \mathcal{F}_e\big[\nu_e^{input}(k) + \nu_{aff}(k), \nu_i(k)\big] - \nu_e(k) \\
T\frac{d\nu_i(k)}{dt} &= \mathcal{F}_i\big[\nu_e^{input}(k) + \nu_{aff}(k), \nu_i(k)\big] - \nu_i(k) \\
\tau_w\frac{dW(k)}{dt} &= -W(k) + b\tau_w\nu_e(k) \\
&\quad + a(\mu_V(\nu_e(k), \nu_i(k), W(k)) - E_L),
\end{aligned}
\tag{5}
$$

where $\nu_e(k)$ and $\nu_i(k)$ are the excitatory and inhibitory population firing rates at site $k$, respectively, $W(k)$ is the level of adaptation of the population, and $\nu_e^{input}(k)$ is the excitatory synaptic input. The latter is given as follows:

$$
\nu_e^{input}(k) = \nu_{drive}(k) + \sum_j C_{jk}\,\nu_e(j, t - \|j - k\|/v_c)
\tag{6}
$$

where the sum runs over all nodes $j$ sending excitatory connections to node $k$, and $C_{jk}$ is the strength of the connection from $j$ to $k$ (and is equal to 1 for $j = k$). Note that $\nu_e(j, t - \|j - k\|/v_c)$ is the activity of the excitatory population at node $k$ at time $t - \|j - k\|/v_c$ to account for the delay of axonal propagation. Here, $\|j - k\|$ is the distance between nodes $j$ and $k$ and $v_c$ is the axonal propagation speed.

As detailed in a previous study [14], the spike-frequency adaptation parameter in the model can be linked to the neuromodulatory drive. From a biological perspective, during wakefulness, increased concentrations of neuromodulators—such as acetylcholine, NE,

and 5HT—diminish spike-frequency adaptation by down-regulating various K+ channels, which results in a prolonged depolarization of the neurons and enables the emergence of asynchronous and irregular firing [20]. On the contrary, during unconsciousness, the lower concentrations of neuromodulators leave K+ channels open. The adaptation build-up and its subsequent decay results in synchronous hyperpolarization and depolarization of the neuron's membrane potential, which, in turn, leads to the generation of slow-wave dynamics. *b* controls the strength of spike-frequency adaptation, so that augmenting *b* corresponds to a reduced neuromodulatory drive and switches the activity from asynchronous to slow waves (see details in [8,13,14]).

### 2.4. Connectomes for the Three Species

The mouse connectome used here is a parcellation comprising 98 regions. The connectivity matrix was created using the Allen Connectivity Builder, which uses high-resolution anterogade tract-tracing data provided by the Allen Institute of Brain Science. The experiments concern source regions only in the right hemisphere; therefore, the left hemisphere is built as a mirror image of the right (data are available at https://zenodo.org/records/8331 301, accessed on 17 January 2023; for more information, see [21]).

The macaque structural connectivity matrix consists of 82 nodes and was generated through a synthesis of axonal tract-tracing and diffusion-weighted imaging data, resulting in a directed and weighted whole-cortex macaque connectome (data available at https://zenodo.org/records/7011292, accessed on 17 January 2023; for more information, see [22]).

The human connectome includes 68 nodes, for which the connection was based on human tractography methods from the Berlin empirical data processing pipeline [23]. Diffusion-weighted imaging does not provide information on fiber tract directionality, but this information can be derived from tracer studies on macaques and subsequently mapped on the human brain (data available at https://zenodo.org/records/4263723, accessed on 17 January 2023).

### 2.5. Integration in The Virtual Brain

The integration of networks of mean-field models was done for each species using The Virtual Brain (TVB) simulator (https://www.thevirtualbrain.org/tvb, accessed on 17 January 2023). For the mouse brain, we used "The Virtual Mouse Brain" [21]; the monkey brain model was provided by the macaque TVB [22]; and the TVB model of the human brain was from [24]. These publications should be consulted for details about the connectivity used. The particular implementation of the human TVB model was that given in [14], where more details can be obtained.

### 2.6. Analysis

To quantify the amount of synchrony in the TVB model, we computed the phase-lag index (PLI) for each pair of nodes, averaged over the simulation time. The Hilbert transform is first computed to extract the phase $\psi(t)$ of the time series. From there, the PLI is given as follows:

$$\text{PLI} \equiv |< sign(\psi_i(t) - \psi_j(t)) >|, \qquad (7)$$

where $< \cdot >$ denotes averaging over time for nodes *i* and *j* [25]. Note that the PLI takes values between 0 (random phase relations or perfect asynchrony) and 1 (perfect phase locking). In this work, we report the mean PLI over all time epochs for the excitatory and inhibitory firing rates of each region pair for each adaptation value.

## 3. Results

We start by showing the behavior of the mean-field model, then we show its integration in brain-wide models of mouse, monkey, and human cerebral cortexes.

Figure 2 shows the spiking network model of AdEx neurons, comprising RS (blue) and FS (red) cells, as detailed in previous studies [6,8]. This network can generate asynchronous irregular (AI) activity states (Figure 2A, top), as well as slow synchronized activity in

the form of up–down states (Figure 2A, bottom). The transition from these two states is obtained by the strengthening or weakening of the spike-frequency adaptation in RS cells (parameter *b* in the model). The mean rates of activity of these two states are shown in Figure 2B. The mean-field model of this AdEx network was derived in two previous publications [6,8] and is shown for these two states in Figure 2C.

As shown previously [13,14], the AdEx mean-field model was implemented in The Virtual Brain (TVB) platform. This platform consists of a Python-based simulation environment [24] that allows the user to create a network of mean-field models (or, more generally, neural mass models) constrained by connectivity data extracted from a given connectome. The TVB environment can also generate a number of neural signals and be linked to neuro-imaging [26]. Integrating the AdEx mean-field model in TVB, leading to the "TVB-AdEx" model [13], has a double advantage. First, the mean-field model is biologically informed, so it has physically interpretable parameters (such as conductances and synaptic receptor kinetics) that can be changed and directly compared to experiments. Second, the AdEx mean-field model can generate states of activity that can be asynchronous or synchronized (see Figure 2), which is not the case with other neural mass models. This behavior is remarkably robust within the large parameter space of the model, as investigated in detail previously [27] and illustrated in Figure 3.

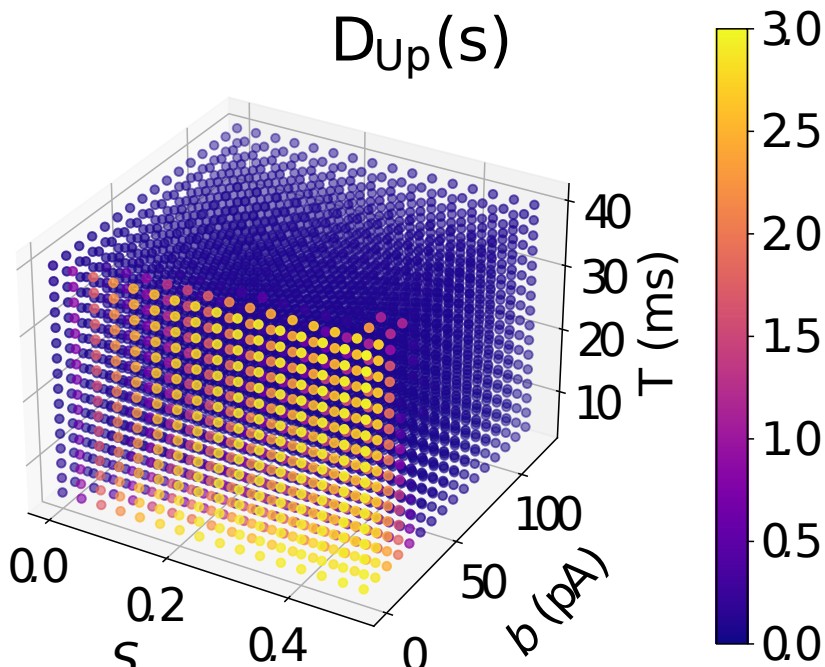

**Figure 3.** Robustness of asynchronous and synchronized slow oscillatory states in the human TVB-AdEx model. The 3D graph represents the TVB model's behavior in a space spanned by three parameters: the connection strength *S*, the level of adaptation *b*, and the time scale *T*. The color indicates the duration of the up states $D_{up}$ (large durations indicate that the dynamics are sustained and asynchronous). Adapted from [27].

The behavior of the TVB-AdEx model is illustrated in Figure 4 for mouse, monkey, and human brains. In all three species, the asynchronous activity of the mean-field model leads to global dynamics that are also asynchronous (Figure 4, middle panels). While there were some variations in the absolute amplitudes of the firing rates, the three models displayed consistent asynchronous dynamics. For higher levels of adaptation, the three models displayed slow-wave activity, as in the mean-field model. However, in this case, the individual nodes synchronized their slow oscillation, which generated globally synchronized slow-wave activity across large brain regions. In the case of the human brain, a

grid scan of parameters showed that the level of synchrony depends on the parameters and, in particular, on the strength of the long-range connectivity [14].

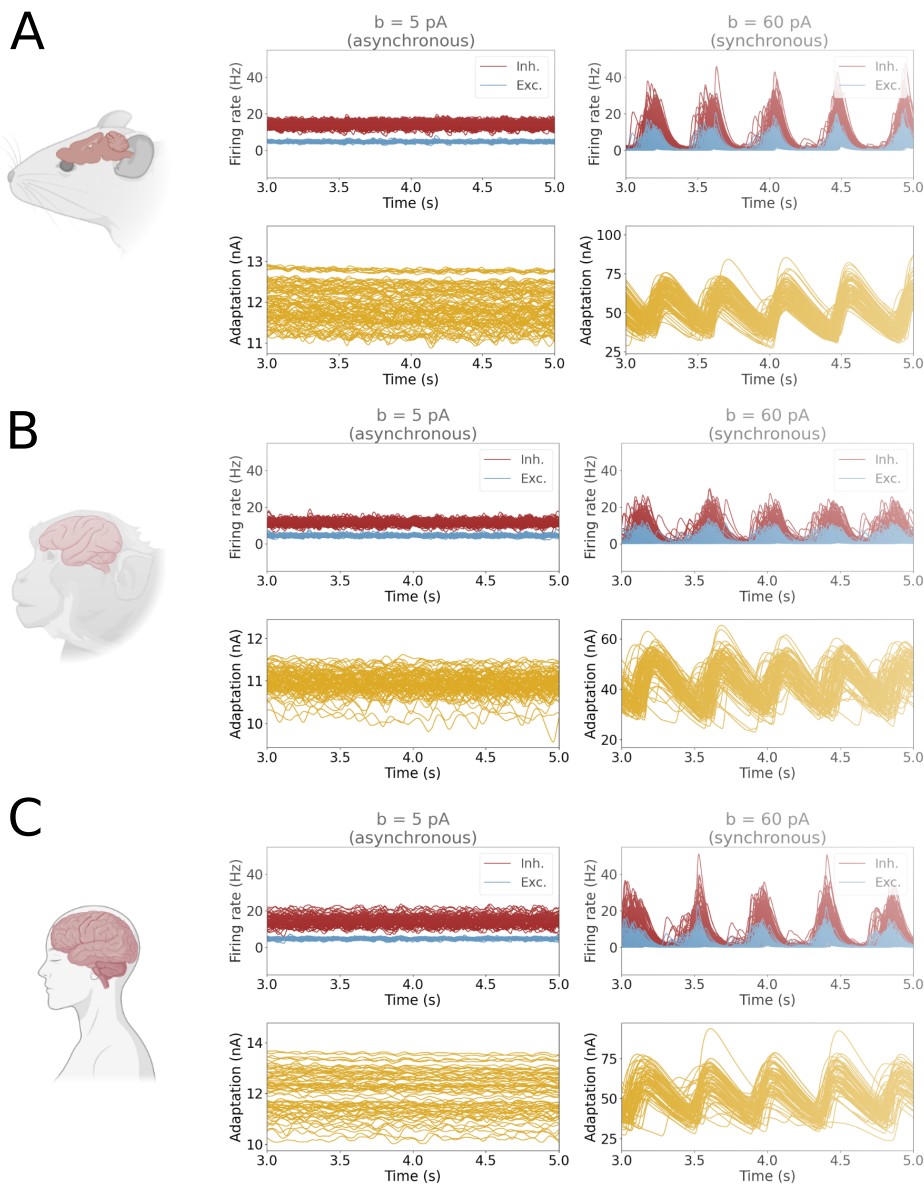

**Figure 4.** Emergent asynchronous and synchronized slow-wave dynamics in mouse, macaque, and human brain models. (**A**) Mouse brain, (**B**) macaque brain, and (**C**) human brain. In each case, the middle panels show the asynchronous activity resulting from low levels of spike-frequency adaptation (*b* = 5 pA), while the right panels show the slow-wave dynamics obtained for higher levels of spike-frequency adaptation (*b* = 60 pA). The color code is blue for excitatory firing rates, red for inhibitory firing rates, and yellow for mean adaptation. Panel C was adapted from [14].

A comparison of the two states across the models of the three species is further shown in Figure 5. The transition from asynchronous to slow-wave dynamics occurs at about the same value of adaptation parameter *b* (Figure 5A). Interestingly, the duration of the down states is predicted to be similar between species. One can also see that the down state duration increases with *b*, and thus the slow-wave frequency also decreases proportionally to the level of adaptation. The level of synchrony of the activities is shown in Figure 5B,C, as measured by the phase-lag index (PLI), for excitatory and inhibitory population activities, respectively. The synchrony increases for excitatory cells as a function of adaptation parameter *b* (Figure 5B). Interestingly, one can see that the level of synchrony

is systematically higher for the mouse brain, intermediate for the human brain, and lower for the monkey brain. Because the mouse brain is considerably smaller, the axonal delays are also smaller than in the other two models. To test its effect on synchrony, we matched the delay distributions of the three species. The axonal propagation speed of the mouse and macaque brains was decreased, resulting in increased delays in these two species that they are similar to the delay distribution in humans. In this case, as shown in Figure 5C, the level of synchrony of the mouse brain diminishes and roughly matches that of the human brain. However, the macaque brain still displays a lower level of synchrony, which may be due to sparser connectivity. This prediction of the model should be tested by conducting appropriate experiments.

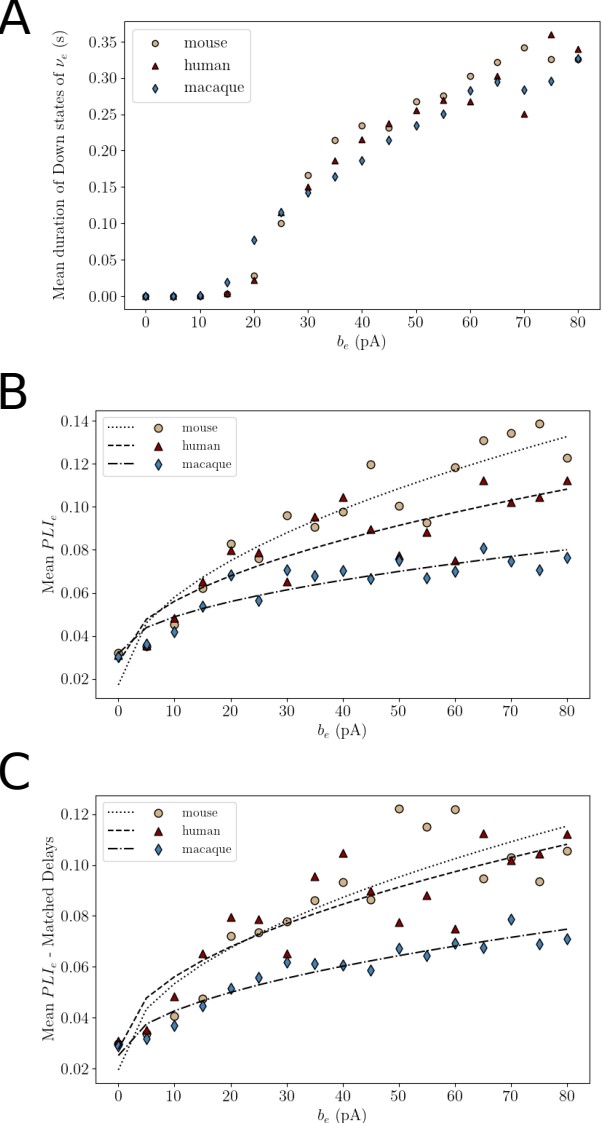

**Figure 5.** Comparison of the asynchronous and synchronized slow-wave dynamics in mouse, macaque, and human brain models. (**A**) Transition between asynchronous and slow-wave dynamics measured by the mean duration of down states. (**B**) Mean phase-lag index (PLI) for excitatory neuron activities for the three species. (**C**) Same as B, but with the connection delays matched between the three connectomes.

## 4. Discussion

In this paper, we present the integration of AdEx mean-field models into whole-cortex simulations for three species: mice, monkeys, and humans. The chosen model is

a second-order mean-field model, grounded in biophysically plausible spiking models, built using a bottom-up approach. The model is able to reproduce the asynchronous and slow oscillatory states of neural networks and therefore permits us to evaluate the emergence of synchronized slow waves at the whole-brain level across three distinct species. For each species, the simulations exhibit the main features of the TVB-AdEx model, as found earlier [13,14]. In particular, the three models exhibit the same emerging property found in human brain simulations, namely that when the individual AdEx nodes are set to asynchronous mode, the ensemble of nodes remains asynchronous. This mode of activity is consistent with the "desynchronized" activity classically found in the awake human brain [1,2]; this is also seen in both human and monkey brains in microelectrode recordings, which also display asynchronous irregular states [28]. Similarly, when the AdEx nodes are in the up–down state, the activity across the whole cortex synchronizes into a slow oscillatory mode. Moreover, this slow oscillatory mode corresponds to the up–down states seen in single units in microelectrode recordings of humans and monkeys during slow-wave sleep [28,29]. Microelectrode recordings in mice also display the same two states—asynchronous irregular dynamics during wakefulness and up–down state dynamics during slow-wave sleep [30]; the same can be seen in cats [31].

We believe that the simulations explored here, and in those for which we provide Python code [32], constitute a very useful tool for the community of computational neuroscientists interested in simulating whole-brain dynamics in mice, monkeys, and humans. We must note here that the codes we provide are run within the TVB environment [24]. In addition, we have also implemented versions of these models that can be run online, on the EBRAINS platform. This will allow users to run the models and change the parameters without even installing TVB, which may also be a useful tool.

Finally, we would like to mention a few possible limitations and extensions to the models explored here. The connectomes used in our study have been derived from different experimental methods (axonal tract-tracing, diffusion weighted images, or a combination of both) and they may exhibit discrepancies in terms of spatial resolution, sensitivity, and accuracy due to inherent differences in the underlying data acquisition and processing procedures. As a direct implication, the connectomes lack consistent directionality (since diffusion-weighted imaging does not provide this information), which may have an effect on the connectivity patterns and their emergent dynamics. It is also important to keep in mind that these are purely cortical models, ignoring the contribution of subcortical structures in the generation of slow-wave activity, with rather coarse brain parcellations that include less than a hundred regions for each species.

A first suggested extension to this study is thus to complete the present model, which is only cortical, by including other regions, such as the thalamus, hippocampus, basal ganglia, or cerebellum. Integrating these regions will require the design of mean-field models specific to each brain region, which was achieved recently for the cerebellum [33] and thalamus [34], and is in progress for other brain regions. The models would also benefit from the use of high-resolution parcellations that take into consideration local circuits and dynamics. A second extension is to study other states and other transitions than the ones explored here. For example, oscillatory states (such as beta or gamma) may occur as responses to sensory stimulation, slow-wave states occurring under anesthesia, or pathological states such as coma or minimal consciousness. To model such states, the procedure is the same as that outlined here: start with a spiking network model displaying these new states, then design a mean-field model that can capture this effect, and—finally—integrate the mean-field model on a large scale in TVB. We believe that these are interesting directions to explore and that the models proposed here will provide useful tools for this exploration.

**Author Contributions:** Conceptualization, J.S.G. and A.D.; methodology, J.S.G., M.S. and L.K.; analysis and preparation of the figures, M.S.; writing of the initial manuscript, A.D.; supervision, project administration and funding acquisition, A.D. All authors have discussed the results and agreed to the published version of the manuscript.

**Funding:** Research supported by the CNRS and the European Union (Human Brain Project H2020-785907, H2020-945539).

**Institutional Review Board Statement:** Not applicable.

**Data Availability Statement:** The source code of all simulations shown in this article is available online. A repository is available in *Zenodo* [32], and the code can be run online using the simulation capabilities offered by the Human Brain Project's EBRAINS neuroscience research infrastructure (https://ebrains.eu and https://ebrains.eu/service/the-virtual-brain, accessed on 17 January 2023).

**Acknowledgments:** We would like to thank Bahar Hazal Yalçinkaya, Trang-Anh Nghiem, David Aquilue, Kevin Ancourt, and Viktor Jirsa for their help, support, and discussions.

**Conflicts of Interest:** The authors declare no conflicts of interest.

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
