# Peer review of "Asynchronous and Slow-Wave Oscillatory States in Connectome-Based Models of Mouse, Monkey and Human Cerebral Cortex"

_applsci, doi:10.3390/app14031063_

Round 1
Reviewer 1 Report
Comments and Suggestions for Authors
This study investigated the emerging properties (asynchronous and slow-wave oscillatory states) in the Adaptive Exponential mean-field models that were integrated into connectome models. This method has already been published by the authors (Goldman et al., 2020; 2022) and was applied in the present study to the connectome date of mouse and macaque monkey. They found that the mouse brain displays a higher synchrony of slow waves than monkey and human brains.
* The main concern is that it is unclear what the connectomes contribute. Could this phenomenon also appear even from a random topology? Results from randomly shuffling connectomes are needed.
* Why was the synchrony of slow-waves in the mouse model higher than those in the human and monkey models? What are the differences between those connectomes? Please discuss these at least.
* Please explain more about the connectome date. How were these connectomes estimated? How many nodes are in each connectome? Are these directions for these connectomes?
* What does parameter b mean physiologically? How does the parameter switch between sleep and wave states?
Reviewer 2 Report
Comments and Suggestions for Authors
This paper draws upon previous research from the same team, where the TVB-AdEx whole-brain model is used to the transition between wakefulness and sleep in humans, mice and macaques. This is done through the variation of a control parameter of the model, namely the adaptation level b, between a value of 5 pA to 60 pA. The results for the human cohort have been presented before [Goldman et al 2023], but they are novel for the other two species.
First of all, I would like to commend the authors for exploring this line of research and extending previous results in new directions. I have been following the developments in this area quite closely, and it is always pleasure to see research connecting biology and mathematics in such an interesting way.
Overall, I think the paper is quite clear and easy to follow, provided some familiarity with the subject. The only lacking aspect, in my humble opinion, is the mention of a previous experiment that is referred to but not shown here: The paper mentions a grid scan of parameters showing that the level of synchrony depends on the model parameters and on the strength of the long-range connectivity. I do not know the space limitations imposed by this journal, but I think the spare space left could be well used to add a simple example of this search, so readers do not have to search for a different paper to understand this one.
Finally, I would like to remind the authors not to forget to fill in the reference to the code at Zenodo, as now it looks... empty!
In any case, let me emphasize I really enjoyed reading this paper and I am looking forward seeing it published.
Reviewer 3 Report
Comments and Suggestions for Authors
Abstract:
The abstract provides a comprehensive overview of the study. However, it would benefit from a brief mention of the key findings regarding the differences in synchrony between the species. This would help to succinctly communicate the study's significance.
Materials & Methods:
Modeling Approach:
The description of the spiking network model, mean-field models, and their integration into The Virtual Brain (TVB) is well articulated. It would be beneficial to include a rationale for the choice of these specific models and their relevance to the study’s objectives.
AdEx Network and Mean-field Models:
The manuscript does a good job of detailing the technical aspects of the models. A brief discussion on the biological implications or relevance of these parameters (e.g., adaptation, connectivity strength) in real brain activity would enhance the reader's understanding.
Connectomes for Three Species:
The sourcing and rationale behind the choice of specific connectomes for the mouse, macaque, and human models are clearly stated. It would be interesting to discuss the limitations or challenges of using these connectomes, if any.
Results:
Behavior of Mean-Field Model and Integration in TVB:
The results section effectively demonstrates the application of the models. However, the manuscript would benefit from a more detailed analysis or discussion on how the findings relate to known physiological behaviors in these species during wakefulness and sleep.
Author Response
(see details in attached PDF file)

Round 2
Reviewer 1 Report
Comments and Suggestions for Authors
The authors’ reply to my major comment: “We tried reshuffling the connections and it had only a minimal effect on the synchrony.” Does this mean that synchronized node activity as in the connectome models was observed even in randomly connected networks? If so, the connectome did not contribute to the emergent (a)synchronous dynamics. This result is contrary to this paper’s main claim that the connectome models are useful for studying simulations of brain activity. The authors should address functional phenomena that emerge from the connectome models, not from random networks. Therefore, I do not recommend the publication of this paper.
Author Response
The reviewer misunderstood our point. On our previous answer, by shuffling the connections we meant the shuffling of the connection weights while retaining the architecture of the tract lengths (and therefore of the delays). We show that indeed, the exact target of the connections are not relevant to the synchrony, but we show (with the data added at the previous round) that it is the delays that matter. The mouse is more synchronous because it has smaller delays between brain regions. We are sorry that the Reviewer did not understand our point, and we have tried to make it more explicit in the paper.